# Exosomes from *EGFR*-Mutated Adenocarcinoma Induce a Hybrid EMT and MMP9-Dependant Tumor Invasion

**DOI:** 10.3390/cancers14153776

**Published:** 2022-08-03

**Authors:** Amina Jouida, Marissa O’Callaghan, Cormac Mc Carthy, Aurelie Fabre, Parthiban Nadarajan, Michael P. Keane

**Affiliations:** 1School of Medicine, University College Dublin, D14 E099 Dublin, Ireland; amina.jouida@ucd.ie (A.J.); marissa.ocallaghan1@ucd.ie (M.O.); cormac.mccarthy@ucd.ie (C.M.C.); afabre@svhg.ie (A.F.); sonadarp2@yahoo.ie (P.N.); 2Department of Respiratory Medicine, St. Vincent’s University Hospital, D04 T6F4 Dublin, Ireland

**Keywords:** exosome, *EGFR*, partial EMT, lung cancer

## Abstract

**Simple Summary:**

Exosomes, a class of extra cellular nano-sized vesicles (EVs), and their contents have gained attention as potential sources of information on tumor detection and regulatory drivers of tumor progression and metastasis. We report that exosomes from serum of patients with Epidermal Growth Factor Receptor (*EGFR*) -mutated non-small cell lung cancer (NSCLC) induce invasion by promoting hybrid EMT. We depict exosomes as valuable actors of metastasis formation and establishment of pre-metastatic niche. Patients with an *EGFR* mutation are prone to metastatic recurrence. Exosomes should therefore be strongly considered as key players in cancer relapse but also as major actors in the establishment of the pre-metastatic niche.

**Abstract:**

Exosomes, a class of extra cellular nano-sized vesicles (EVs), and their contents have gained attention as potential sources of information on tumor detection and regulatory drivers of tumor progression and metastasis. The effect of exosomes isolated from patients with an Epidermal Growth Factor Receptor (*EGFR*)-mutated adenocarcinoma on the promotion of epithelial–mesenchymal transition (EMT) and invasion were examined. Exosomes derived from serum of patients with *EGFR*-mutated non-small cell lung cancer (NSCLC) mediate the activation of the Phosphoinositide 3-kinase (PI3K)/AKT/ mammalian target of rapamycin (mTOR) pathway and induce an invasion through the up-regulation of matrix metalloproteinase-9 (MMP-9) in A549 cells. We observed a significant increase in the expression of vimentin, a mesenchymal marker, while retaining the epithelial characteristics, as evidenced by the unaltered levels of *E-cadherin* and *Epithelial cell adhesion molecule (EPCAM*). We also observed an increase of *nuclear factor erythroid 2-related factor 2 (NFR2)* and *P-cadherin* expression, markers of hybrid EMT. Exosomes derived from *EGFR*-mutated adenocarcinoma serum could be a potential mediator of hybrid EMT and tumor invasion. Understanding how cancerous cells communicate and interact with their environment via exosomes will improve our understanding of lung cancer progression and metastasis formation.

## 1. Introduction

Exosomes are nano-sized vesicles (between 30 and 150 nm) with an endocytic origin. They are found in all biological fluids. They are widely recognized for their role in mediating cell-to-cell communication and function as intercellular carriers for various biomolecules, including proteins, RNAs, microRNA, and DNA [1,2,3,4]. Recent studies document that exosomes are released at a significantly higher rate by tumor cells, and they play critical roles in several early and late events related to tumor growth and metastasis [5,6,7]. Thus, circulating exosomes are emerging as a new model of ‘liquid biopsy’ for non-invasive cancer diagnosis. Nevertheless, new technologies allowing better detection and characterization of specific exosomal markers reflecting their subtype and functions are still needed for an effective clinical application of exosomes [8]. Emerging evidence in the lung cancer context has shown that exosomes are associated with EMT (epithelial-to-mesenchymal transition) and carry several EMT signature molecules [9]. This process provides a remarkable example of cellular plasticity during cancer dissemination [10]. During EMT, epithelial cells lose all relics of their epithelial origin and acquire a fully mesenchymal phenotype. Interestingly, this transition is not a binary process but instead, cells often present a range of epithelial/mesenchymal phenotype(s), where cells co-express both epithelial and mesenchymal markers, collectively known as partial or hybrid EMT states. Such hybrid cells can move collectively as clusters, which is thought to enhance their invasive properties [11].

Lung cancer remains the leading cause of cancer death worldwide and has a poor prognosis (five-year survival rate 10%). Two distinct morphological classifications can be identified in lung cancer: small cell lung cancer (SCLC), which accounts for approximately 15% of the total number of cases, and non-small cell lung cancer (NSCLC), which accounts for around 85%. Three main subtypes of NSCLC can be identified: adenocarcinoma (accounting for about 50% of all NSCLC), squamous cell carcinoma (which accounts for approximately 30%) and large-cell undifferentiated carcinoma (comprising about 15%) [12]. In order to provide diagnosis at an early stage, targeted therapy, and the development of medications, it is critical to grasp the biological mechanisms inherent to the development of tumors, as well as possible cancer biomarkers expression. The understanding of genomic alterations in cancer, thanks to extensive collaborative work, have enabled the identification of possible diagnostic and therapeutic targets, one of which is the epidermal growth factor receptor (*EGFR*). In lung adenocarcinoma, *EGFR* mutation has a predominance that ranges from 10% to 78% and vary significantly depending on ethnic origin and geographic location [13]. Recent findings indicate that *EGFR*-mutated cancers are correlated with increased occurrence of diffuse lung metastases [14]. Current guidelines highly recommend molecular testing of lung cancer patients [15]. However, in up to 20% of patients, tissue biopsies are either inconclusive or inaccessible due to the lack of sufficient neoplastic tissue, or because it is technically not feasible [16]. Consequently, liquid biopsy has the potential to become a complementary/alternative tool for diagnosis and prognosis to conventional tissue biopsy [17].

In lung cancer with an *EGFR* mutations, an increasing body of evidence has shown that exosomes can provide important information regarding tumor initiation, progression and metastasis formation [18]. One of the key drivers of metastasis initiation is EMT. Although EMT has been extensively studied in the context of cancer, the role of exosomes secreted by *EGFR*-mutated tumors in the process of EMT, partial EMT and collective migration remains under-investigated. For these reasons, we decided to investigate the role of exosomes from non-small cell lung cancer patients, particularly focusing on exosomes from *EGFR*-mutated patients. Our findings indicate that exosomes derived from *EGFR*-mutated patients might play an important role in driving invasion and metastasis via a Phosphoinositide 3-kinase (PI3K)/AKT/ mammalian target of rapamycin (mTOR)/ matrixmetalloproteinase-9 (MMP-9) pathway.

## 2. Materials and Methods

### 2.1. Serum Samples

Whole blood samples from 15 patients with NSCLC and 5 healthy individuals were collected before treatment at St. Vincent’s University Hospital, Dublin, Ireland. Blood samples were collected in coagulation-promoting tubes (Becton, Dickinson and Company, Franklin Lakes, NJ, USA) and centrifuged at 5000× *g* for 10 min and then at 15,000× *g* for 10 min, at 4 °C. The serum samples were stored at −80 °C until needed. The clinical characteristics of the patients are presented in Appendix A.

### 2.2. Cell Lines

Cell lines were obtained from the American Type Culture Collection (ATCC) (A549: CCL-185, HBE4-E6/E7: CRL-2078, HCC827: CRL-2868). A549 cells were cultured in in Dulbecco’s modified Eagle’s medium (DMEM) supplemented with 10% fetal bovine serum (FBS), 50 U/mL penicillin, and 50 μg/mL streptomycin in a 5% CO_2_ humidified incubator at 37 °C. HCC827 cells were cultured in Roswell Park Memorial Institute (RPMI)-1640 medium supplemented with 10% FBS. HBE4-E6/E7 cells were cultured in Keratinocyte-SFM supplemented with 0.2 ng/mL EGF and 25 µg/mL bovine pituitary extract.

### 2.3. Isolation of Exosomes from Serum and Cell Supernatant by Differential Centrifugation

HCC827 cells were cultured in a conventional culture medium for 48 h. The medium was then replaced with RPMI-1640 supplemented with 2% (*v/v*) exosome-depleted FBS (System Biosciences, LLC, Palo Alto, CA, USA) until 90% confluence. Exosomes were isolated from the cell culture supernatant by differential centrifugation. Samples were centrifuged at 300× *g* for 10 min then 2000× *g* for 10 min and 10,000× *g* for 30 min at 4 °C to remove cells and cell debris. Supernatant was collected into ultracentrifuge tubes and centrifuged in an Optima MAX-XP ultracentrifuge with an MLA-130 rotor (Beckman Coulter, Jersey City, NJ, USA) at 100,000× *g* for 1 h at 4 °C. The pellets were washed with phosphate-buffered saline, ultracentrifuged again, and resuspended in PBS. Serum exosomes were isolated using the same differential centrifugation protocol. The protein concentration was determined using a bicinchoninic acid (BCA) assay kit. Exosome preparations were conserved at −80 °C for later use. The morphological characteristics of exosomes were observed under a transmission electron microscope (TEM, Philips, The Netherlands). The size distribution of exosomes was measured by NanoSight NS500 System (Nanosight Ltd., Malvern, UK). Expression levels of representative exosome-related markers Calnexin, CD81, and CD63 were detected by Western blot (see below).

### 2.4. Western Blot Analysis

Total cellular protein lysates from A549 and HBE4 E6/E7 and serum exosomal protein were obtained using RIPA buffer supplemented with proteinase inhibitors, according to manufacturer’s protocols, and centrifuged at 14,000× *g* for 10 min at 4 °C. Protein concentrations were then determined using a BCA assay. Proteins (30 µg for exosomes and 10 µg for cell lysates) were separated by 10% SDS-PAGE gel and transferred to the polyvinylidene fluoride (PVDF) membrane. After being blocked with 5% skimmed milk at room temperature (RT) for 2 h, the membranes were incubated with the primary antibodies at 4 °C overnight. The primary antibodies used for the Western blot were as follows: CD63 (1:500; Abcam, Cambridge, UK, Cat#: ab59479), CD81 (1:500; Santa Cruz Biotechnology, Dallas, TX, USA, Cat#: sc-23962), Calnexin (1:1000; Santa Cruz Biotechnology, Dallas, TX, USA, Cat#: sc-80645), E-cadherin (1:1000; Cell Signaling, Danvers, MA, USA, Cat#: mAb3195), vimentin (1:1000; Santa Cruz Biotechnology, Dallas, TX, USA, Cat#: sc-373717) and GAPDH (1:10,000; Millipore, Burlington, MA, USA, Cat# mAb374). Following 1 h incubation with goat anti-mouse or goat anti-rabbit secondary antibody (ThermoFisher Scientific, Waltham, MA, USA, Cat#: 35569 and SA-535521), the signal was visualized using an LI-COR-Odyssey infrared scanner (LI-COR, Lincoln, NE, USA). Analysis was conducted using Image Studio Lite Version 5.2 software (supplied by LI-COR, Lincoln, NE, USA). GAPDH was utilized as a control.

### 2.5. Phospho-Kinase Array

The Human Phospho-Kinase Array (RD systems, Abingdon, UK, Cat#: ARY003B) was used for analyzing phosphorylation of 43 kinases according to the manufacturer’s instructions. A549 cells were treated with 50 µg of exo-HCC827 for 48 h then harvested and lysed. Cell lysates containing a total of 500 μg protein were incubated overnight with the antibody array, and the bound phospho-kinases were detected with anti-phospho-tyrosine antibody-horseradish peroxidase (HRP) (1:2000). The spot signals were quantified using ImageJ software and normalized to the internal reference spots first.

### 2.6. Transmission Electron Microscopy of Exosomes

To identify the presence of exosomes, the samples were analyzed using transmission electron microscopy (TEM). In brief, exosomes were dropped onto a formvar carbon-coated copper grid and then fixed by 2.5% glutaraldehyde in cold DPBS for 10 min. After a wash, the samples were contrasted with 2% uranyl acetate for 15 min. The samples were then embedded by adding methyl cellulose-UA for 10 min on ice. Excess cellulose was removed, and the samples were dried for permanent preservation. Finally, the preparation was examined with a transmission electron microscope.

### 2.7. Nanotracking Analysis

Analyses were performed using NanoSight NS300 (Malvern, UK). The exosomes were diluted 100 times in PBS and added to the NanoSight sample chamber. Experimental videos were analyzed using nanoparticle tracking analysis (NTA) 2.3 build 17 software (Malvern) after capturing them in the script control mode (15 videos of 60 s per measurement) using a 1 mL injection syringe (Becton Dickinson, Franklin Lakes, NJ, USA).

### 2.8. Immunofluorescence

EMT gene expression was analyzed by immunofluorescence. Rabbit anti-E-cadherin antibody (1:50; Cell Signaling, Danvers, MA, USA Cat#: mAb3195) and Mouse anti-Vimentin antibody (1:200; Santa Cruz Biotechnology, Dallas, TX, USA, Cat#: sc-373717) were used. Cell samples were first fixed with 4% paraformaldehyde (PFA) for 10 min and then incubated with 5% BSA for 30 min. After blocking non-specific antigens with BSA, cell samples were incubated with the abovementioned antibodies overnight at 4 °C in a humidified chamber. The cells were then incubated with donkey anti-rabbit IgG Alexa Fluor 594 (1:500; ThermoFisher Scientific, Waltham, MA, USA, Cat#: A32754) and donkey anti-mouse IgG Alexa Fluor 488 (1:500; ThermoFisher Scientific, Waltham, MA, USA, Cat#: A32766) for 1 h at room temperature, respectively, after rinsing four times with PBS. Cell nuclei were stained for 10 min with DNA fluorochrome 4′,6-diamidino-2-phenylindole (DAPI) (1:1000; Abcam, Cambridge, UK, Cat#: ab228549). Fluorescence images were obtained under a fluorescent microscope.

### 2.9. Migration and Invasion Assays

Wound-healing assays were used to observe cell migration capacities with and without specific inhibitors: AKTi ½ (Tocris Bioscience, Bristol, UK, Cat#: 5773/10), rapamycin (Tocris Bioscience, Bristol, UK, Cat#: 1292/1) and a specific MMP-9 inhibitor (Abcam, Cambridge, UK, Cat#: ab142180). The cell monolayers were scratched with 200 μL pipette tips. In order to remove cell debris, the main cells were washed three times in PBS and the cells were allowed to migrate for 48 h. Phase-contrast images were successively recorded every 10 min over 48 h with a Nikon Inverted Microscope Incubation Systems Ti-2. The gap size was subsequently analyzed using ImageJ software (http://imageJ.nih.gov/ij/index.html, accessed on 5 September 2018). Experiments were done at least in triplicate. Wound closure (%) was quantified using the equation ((wound healing areas/cell-free area of the initial scratch) × 100%).

Modified Boyden chamber assays (Corning, New York, NY, USA, Cat#: 354481) were used to assess the invasive properties of cells. Cells were resuspended in serum-free medium and placed in the upper compartment of the invasion chamber. The lower compartment was filled with medium containing 10% FBS. The chambers were incubated for 16 h at 37 °C. The two compartments were separated by a porous filter (8 μm pore) coated with Matrigel. The filters were then fixed in methanol and stained with hematoxylin. Quantification of the invasion assay was performed by counting the number of cells on the lower surface of the filters (10 fields at 400-fold magnification). Experiments were done at least in triplicate.

### 2.10. 3D-Organotypic Growth Assays

A549 cells were diluted in exosome-depleted medium supplemented with 5% Cultrex (R&D systems, Abingdon, UK, Cat#: 3432) and seeded onto solidified Cultrex cushions (50 μL/well) contained in 96-well plates (1000 cells/well). Media containing the indicated inhibitors were replaced every 4 days and organoid outgrowth was assessed by phase-contrast microscopy.

### 2.11. Quantitative Real-Time PCR

Total RNA was isolated from cells with the RNeasy Plus Kit (Qiagen, Manchester, UK) according to manufacturer’s instructions, and 500 ng of RNA was reverse transcribed to cDNA as per the manufacturer’s instructions. Quantitative real-time PCR (qRT-PCR) was performed with SYBR Green PCR master mix (Applied Biosystems, Waltham, MA, USA), template cDNA, and primers on an ABI Prism 7900HT Sequence Detector (Applied Biosystems, Waltham, MA, USA). *GAPDH* served as an endogenous control. Relative changes in transcript levels in treated samples compared with controls were expressed by the ΔΔCt method (where Ct is cycle threshold). Primer sequences were as follows in Table 1.

### 2.12. EMT Scoring Method

We calculated EMT score by using RT-qPCR levels of two epithelial (*CDH1* and *EPCAM*) and three mesenchymal (*Vim*, *MMP-2*, and *MMP-9*) markers. We modified the formula used by Chae et al. and Chirshev et al. [19,20]: EMT score=ΔCT (“Mean of mesenchymal markers”)−ΔCT(“mean of epithelial markers”). The higher the EMT score the more epithelial the cells are, and the lower the EMT score the more mesenchymal the cells are.

### 2.13. Statistical Analysis

All experiments were performed independently at least three times. Data are presented as means ± SEM. Data were analyzed with GraphPad Prism version 9.3 (supplied by GraphPad Software, San Diego, CA, USA). An unpaired *t*-test or two-way ANOVA was used to calculate the difference in the expression of the markers, with a *p*-value of <0.05 determined to be statistically significant.

### 2.14. Study Approval

This study received ethics approval from the Ethics and Medical Research Committee of Saint Vincent’s University Hospital.

## 3. Results

### 3.1. Exosomes Derived from Serum of Lung Cancer Patients and HCC827 Cells Were Isolated and Confirmed

The morphological features of the isolated exosomes from HCC827 cell line and from NSCLC patients and healthy donors were identified by transmission electron microscopy (Figure 1A and Figure 2A). The isolated exosomes were found to be approximately 100 nm in diameter. This is confirmed by the results of nanoparticle tracking analysis (Figure 1B and Figure 2B). A panel of exosomal markers was used to check, by Western blotting, the purity of the exosomes. CD63 and CD81 exosomal surface protein markers were detected, while calnexin, used as negative marker, was not detected (Figure 1C and Figure 2C).

### 3.2. Exosomes Derived from Serum of EGFR-Mutated Patients and from EGFR-Mutated Cell Line Promote Extracellular Matrix Degradation and EMT

To understand the role of serum exosomes in patients with *EGFR* mutation, we first analyzed MMP activity by zymography. Exosomes isolated from patients with *EGFR* mutation showed significant increased activity of pro-MMP-9 and MMP-9 compared to exosomes isolated from serum from wild-type adenocarcinoma (ADK WT), squamous cell carcinoma, or healthy donors (Figure 3B,C). Compared to A549 cells treated with serum-derived exosomes from ADK WT, squamous cell carcinoma, and healthy donors, A549 cells treated with exosomes from patients with an *EGFR* mutation showed a significant increase of vimentin and a slight decrease of E-cadherin, canonical markers of EMT [21] (Figure 3D,E). The supernatant of A549 treated with serum-derived exosomes was assessed by zymography. We showed a significant increase in MMP-2, pro-MMP-9, and MMP-9 in A549 cells treated with exosomes from patients with *EGFR* mutation (Figure 3F–H). MMP-2 and MMP-9 activity was also significantly increased in A549 cells treated with exosomes from patients with ADK WT, but interestingly, MMP-9 activity was significantly higher in A549 cells treated with exosomes from patients with *EGFR* mutation.

To further investigate the functionality of exosomes isolated from serum from patients with *EGFR* mutation, we developed a cellular model mimicking their effects in vitro using exosomes from an *EGFR*-mutated cell line: HCC827 harboring *EGFR* Del E746-A750 in exon 19, the most common activating mutation [22]. After exposure to exosomes derived from HCC827 cell line (exo-HCC827), cells were stained for vimentin and E-cadherin using immunofluorescence. A change in cellular architecture was observed, with A549 cells exposed to exo-HCC827 demonstrating a more disorganized structure and acquiring an elongated morphology (Figure 4A). HBE4-E6/E7 cells, an immortalized normal bronchial cell line, were also used to confirm this model (Appendix A). We demonstrated that vimentin expression was significantly up-regulated and E-cadherin expression slightly down-regulated in A549 and HBE4 E6/E7 cells treated with exo-HCC827. This observation was confirmed by Western blot (Figure 4B and Appendix A). MMP activity was assessed by zymography. Cells exposed to exo-HCC827 demonstrated significantly increased activity of MMP-2 and MMP-9 (Figure 4C and Appendix A). The above experiments show that exosomes derived from serum of patients with *EGFR* mutation and from HCC827 cell line promote extracellular matrix degradation and EMT-like processes.

### 3.3. Exosomes Derived from EGFR-Mutated Cell Line Induce Invasion That Is MMP-9 Dependent

To investigate the mechanism underlying extracellular matrix degradation and EMT induced by exosomes derived from HCC827 cell line, we examined the phosphorylation of 43 different kinases using a phosphokinase array. As shown in Figure 5, AKT and mTOR displayed an increased phosphorylation in A549 cells exposed to exo-HCC827. Phosphorylation of AKT in ser473 is a downstream target of PI3K. This result suggests that exosomes from HCC827 activates the PI3K/AKT/mTOR signaling pathway. We then investigated the relationship between MMP-9 expression and this signaling pathway. The increased expression of MMP-9 induced by exo-HCC827 was significantly downregulated with the use of AKT1/2 (AKTi), a specific AKT inhibitor, and rapamycin, an inhibitor of mTOR pathway (Figure 6A). This result suggests that MMP-9 expression induced by exosomes derived from the HCC827 cell line is PI3K/AKT/mTOR dependent.

To further investigate the biological function of exo-HCC827, we evaluated the migratory and invasive properties induced by exosomes in three different complementary models: wound healing assays, allowing the measure of cell migration (see Appendix A); the modified Boyden chamber invasion assay, mimicking basement membrane crossing, an early step of invasion; and a 3D organotypic outgrowth assay, measuring the ability of cells to grow in an anchorage-independent manner thus mimicking the pulmonary microenvironment. Exo-HCC827 significantly increased the migratory and invasive capacities of A549 cells, while AKTi, rapamycin, and ab142180 (a selective MMP-9 inhibitor) decreased this exo-HCC827 invasion and migration induced (Figure 6B,C). A549 cells treated with exo-HCC827 grew robustly in 3D-cultures, while the use of AKTi, rapamycin, and ab142180 inhibited 3D outgrowth significatively (Figure 6D). These results demonstrate the ability of exosomes derived from an *EGFR*-mutated cell line to promote invasion and migration via the activation of the PI3K/AKT/mTOR/MMP-9 pathway.

### 3.4. Exosomes Derived from EGFR-Mutated Cell Line Induce Partial/Hybrid EMT

Our findings suggest that exosomes derived from an *EGFR*-mutated cell line promote EMT. Although, we observed a significant increase of vimentin expression, cells seemed to retain epithelial characteristic, as evidenced by the unaltered levels of *E-cadherin* and *EPCAM* (Appendix A). Therefore, we evaluated by RT-qPCR, the expression of *P-cadherin* and *NRF2 (nuclear factor erythroid 2-related factor 2)*, proposed markers for hybrid/partial-EMT [23,24]. We used TGFβ, a cytokine known to induce complete EMT [25]. We observed an increased expression of *NRF2* and *P-cadherin* in A549 cells treated with exo-HCC827 but not in A549 cells treated with TGFβ (Figure 7A,B). We calculated the EMT score and, as shown in Figure 7C, non-treated A549 cells exhibited an EMT score of 2, but when they were exposed to TGFβ, an inducer of complete EMT, the score was −3.7. When cells were exposed to exo-HCC827, the EMT score was 0.76. These results demonstrate that exosomes derived from an *EGFR*-mutated cell line promote partial EMT.

## 4. Discussion

The contribution of exosomes in lung cancer metastasis is still relatively under investigated. The present study explores the functional effects of exosomes derived from the serum of patients with *EGFR* mutation on invasion and metastasis. It reveals that exosomes derived from serum of patients with *EGFR* mutation mediate the activation of the PI3K/AKT/mTOR pathway and induce an invasion through the up-regulation of MMP-9. The gelatinase MMP-9 plays a crucial role in cancer development and progression [26]. Other studies have reported the presence of MMPs in serum exosomes from different type of cancer, including ovarian, breast, and colorectal [27,28,29]. Here, we confirm for the first time the presence of MMP-9 and its inactive proform in exosomes derived from NSCLC. Interestingly, MMP-9 and pro-MMP-9 are notably elevated in serum-derived exosomes from patients with *EGFR*-mutated adenocarcinoma, and this is independent of stage, gender, age, or smoking history. The expression and activity of MMP-9 is closely associated with invasiveness and metastatic potential due to its ability to degrade gelatin and type IV, V, XI, and XVI collagens during tissue remodeling [30]. To understand the functional implication of exosomes isolated from lung adenocarcinoma patients with an *EGFR* mutation on recipient cells, A549 cells were used. We observed that exosomes isolated from patients with *EGFR*-mutated lung adenocarcinoma promoted extracellular matrix degradation and a significantly increased expression of vimentin in recipient A549 cells. Activation of the PI3K/AKT/mTOR pathway by exosomes derived from an *EGFR*-mutated cell line induced invasion and metastasis through up-regulation of MMP-9. Evidence shows that the PI3K/AKT/mTOR pathway has a significant role in tumor initiation and development and mediates the EMT process [31].

During this process, cells lose their epithelial traits to acquire mesenchymal characteristics and increased motility. It is a dynamic process whereby cells can switch back and forth from one state to another. But this switch is not binary, as cells often exhibit a range of hybrid states; it has been referred to as “partial EMT”. Those hybrid cells can invade collectively via clusters, maintaining their cell-to-cell connections and providing a survival advantage, which is thought to enhance their invasive properties. Their presence has been linked to poor survival and drug resistance in several cancers [32]. The biochemical hallmarks of partial EMT are still under investigation. However, emerging evidence suggests NRF2 and P-cadherin as proposed markers. In the clinical context, overexpression of each of these factors is significantly associated with tumor progression and poor survival [33,34]. NRF2 has been described as a “phenotypic stability factor” due to its ability to prevent completion of EMT and to stabilize partial EMT [35]. P-cadherin overexpression has been described as a disrupter of epithelial adhesions and promoter of a more undifferentiated cell phenotype. Ribeiro and Paredes [24] also describe an increased therapeutic resistance, stemness properties, and a more aggressive phenotype in P-cadherin overexpressing cells. For these reasons, they suggested P-cadherin as a potential marker for hybrid cells. Increased expression of *NRF2* and *P-cadherin* has been observed in cells exposed to exosomes derived from an *EGFR*-mutated cell line but not in cells exposed to TGFβ, a known inducer of complete EMT. The increased expression of *NRF2, P-cadherin,* and *vimentin* by exosomes while co-expressing *E-cadherin* and the exosome-induced invasion proves that exosomes derived from an *EGFR*-mutated cell line promote partial EMT. The extent of the EMT (EMT score) was calculated using gene expression signatures and the findings further corroborate our conclusions. The use of this scoring method is a powerful tool to investigate the dynamic of EMT in cancer progression.

EMT plays a critical role in metastatic recurrence; however, cells that undergo partial EMT appear to pose a higher metastatic risk than cells that undergo complete EMT [36]. Previous studies have shown that *EGFR* mutation in NSCLC is associated with a high risk of metastatic recurrence [37]. Our study provides insight into how exosomes promote invasion and hybrid EMT in *EGFR*-mutated lung adenocarcinoma, most likely through the PI3K/AKT/mTOR pathway. This has potential implications for targeted therapies in the future. Exosomes should therefore be strongly considered as key players in cancer relapse but also as major actor in the establishment of the pre-metastatic niche. Despite the limitation of the small size of the cohort, our study provides strong proof-of-concept for the contribution of exosomes to the establishment of metastasis. Further studies should focus on the exosomes’ cargo to identify the key elements responsible for the effects described in this study. Moreover, we looked at the effect of exosomes derived from *EGFR*-mutated lung adenocarcinoma on epithelial cells. Additional studies examining the effects of these exosomes on other cell types associated with the metastatic process may give a better understanding on how lung cancer spreads.

## 5. Conclusions

This is the first study highlighting the role of exosomes derived from *EGFR*-mutated lung adenocarcinoma (independent of stage, gender, age, or smoking history of patient) in the metastatic process. It was found that they promote partial EMT reprogramming, which orchestrates a pre-metastatic niche establishment, fostering the development of metastasis. Further studies are required to explore and precisely delineate these complex mechanisms.

## Figures and Tables

**Figure 1 cancers-14-03776-f001:**
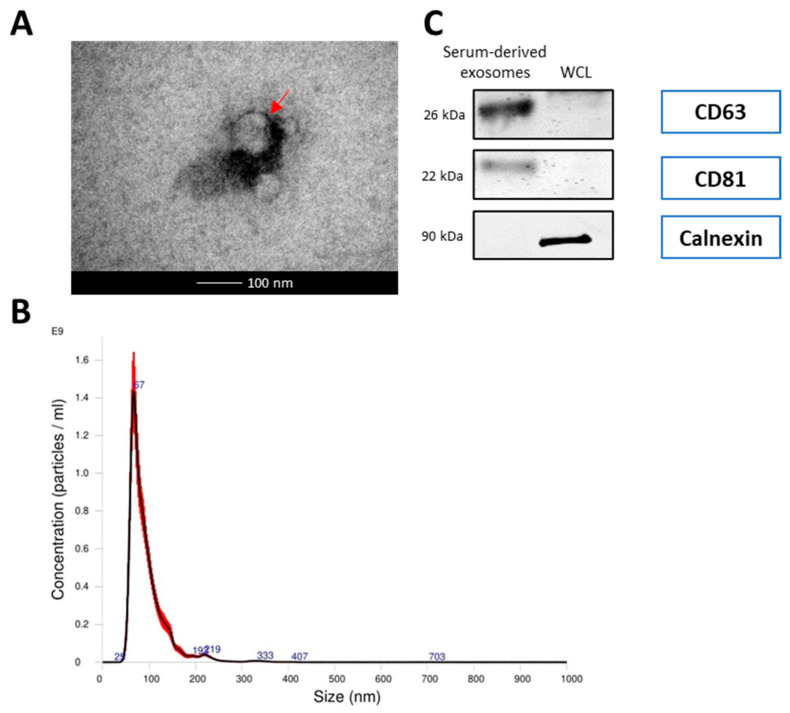
Characterization of serum exosomes (**A**) Transmission electron microscope (TEM) images of exosomes isolated from serum. The bar represents 100 nm. Red arrows indicate exosomes. (**B**) The size distribution of serum exosomes determined by nanoparticle tracking analysis (NTA). (**C**) Western blot analysis of exosome markers (CD63 and CD81) and negative markers (calnexin) in equivalent amounts of protein from serum exosomes and A549 whole cell lysates (WCL) (as a control). (Original whole Western-blot Images and densitometry reading in Figure 1 and Appendix A).

**Figure 2 cancers-14-03776-f002:**
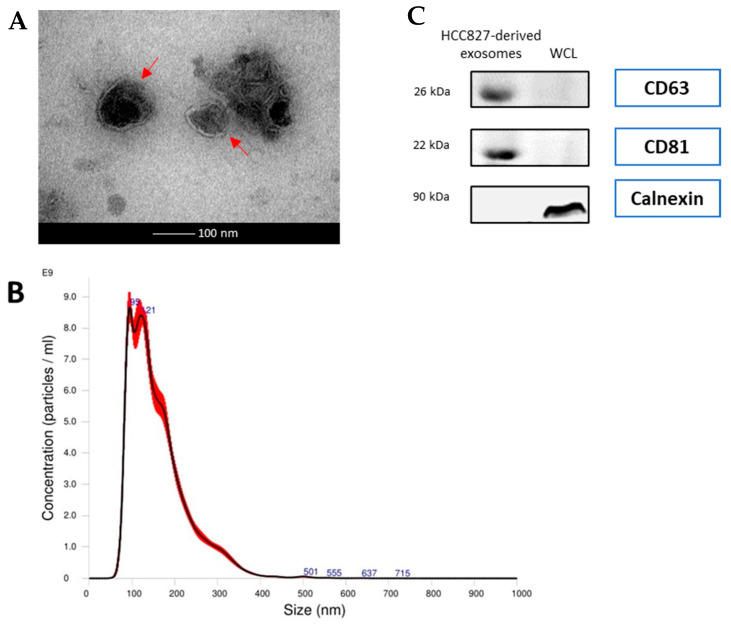
Characterization of exosomes from the HCC827 cell line (**A**) Transmission electron microscope (TEM) images of exosomes isolated from HCC827 cells. The bar represents 100 nm. Red arrows indicate exosomes. (**B**) The size distribution of serum exosomes determined by nanoparticle tracking analysis (NTA). (**C**) Western blot analysis of exosome markers (CD63 and CD81) and negative markers (calnexin) in equivalent amounts of protein from HCC827-derived exosomes and A549 whole cell lysates (WCL) (as a control). (Original whole Western-blot Images and densitometry reading in Figure 2 and Appendix A).

**Figure 3 cancers-14-03776-f003:**
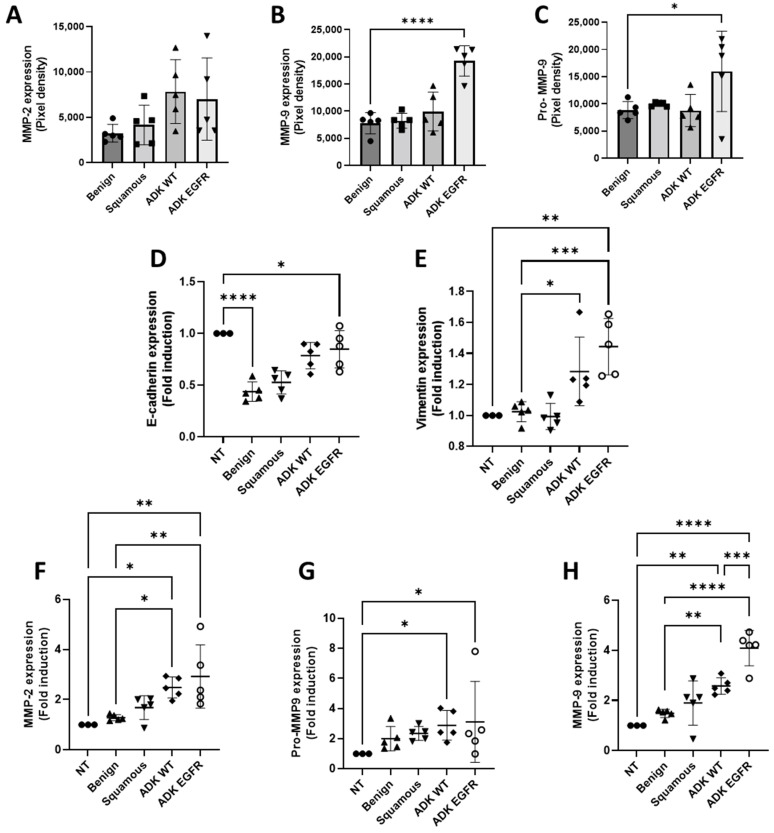
Exosomes from serum of *EGFR*-mutated patients promote extracellular matrix degradation and EMT-like process (**A**–**C**) Quantification analysis of expression of MMPs obtained by zymography in exosomes isolated from serum (mean ± SEM, *n* = 5). (**D**,**E**) Quantification analysis of vimentin and E-cadherin expression obtain by Western blotting of protein extracts of A549 cells treated with exosomes isolated from serum. GAPDH is used as a loading control (mean ± SEM, *n* = 5). (**F**–**H**) Quantification analysis of expression of MMPs obtained by zymography of supernatant of A549 cells treated with exosomes isolated from serum (mean ± SEM, *n* = 5). Significance was set at * *p* < 0.05, ** *p* < 0.01, *** *p* < 0.001 and **** *p* < 0.0001.

**Figure 4 cancers-14-03776-f004:**
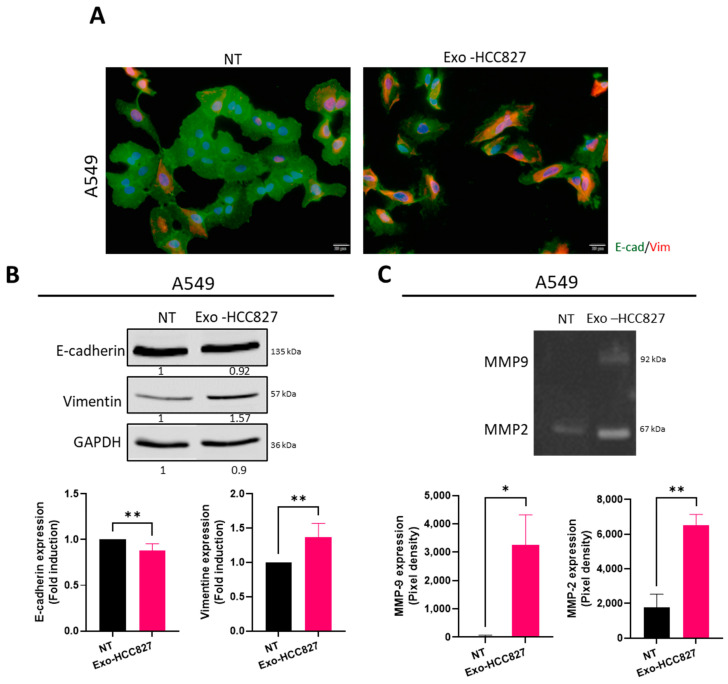
Exosomes derived from the HCC827 cell line promoted extracellular matrix degradation and EMT (**A**) Immunofluorescence detection of vimentin (red) and E-cadherin (green) in A549 cells treated with exosomes isolated from HCC827 supernatant. Nuclei were stained with DAPI (blue). Scale bar = 20 μm. (**B**) Western blot analysis and quantification of vimentin and E-cadherin expression in A549 cells treated with exosomes isolated from HCC827 supernatant. GAPDH was used as a loading control (mean ± SEM, *n* = 4; statistical analysis was performed using unpaired *t*-test). (Original whole Western-blot Images and densitometry reading in Figure 3 and Appendix A). (**C**) Zymography analysis and quantification of A549 cells (**right**) treated with exosomes isolated from HCC827 supernatant (mean ± SEM, *n* = 4; statistical analysis was performed using unpaired *t*-test). Significance was set at * *p* < 0.05 and ** *p* < 0.01.

**Figure 5 cancers-14-03776-f005:**
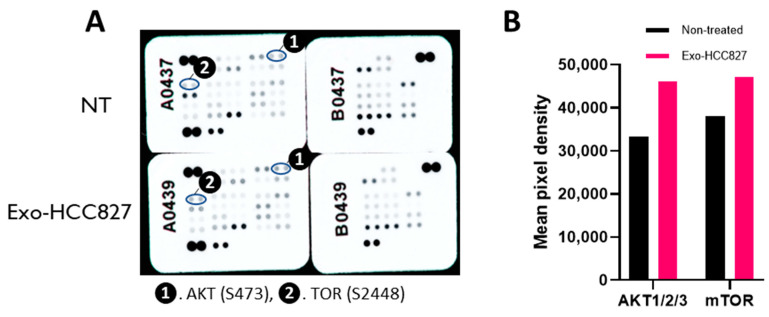
Exosomes from HCC827 cell line activates the PI3K/AKT/mTOR signaling pathway. (**A**) Phosphokinase antibody array blots using lysates of A549 treated with 50 µg of exosomes isolated from HCC827 supernatant. Circled spots indicated positive signal for ❶ AKT (S473) and ❷. TOR (S2448) (**B**) Quantification of the relative levels of the phosphorylation of indicated kinases.

**Figure 6 cancers-14-03776-f006:**
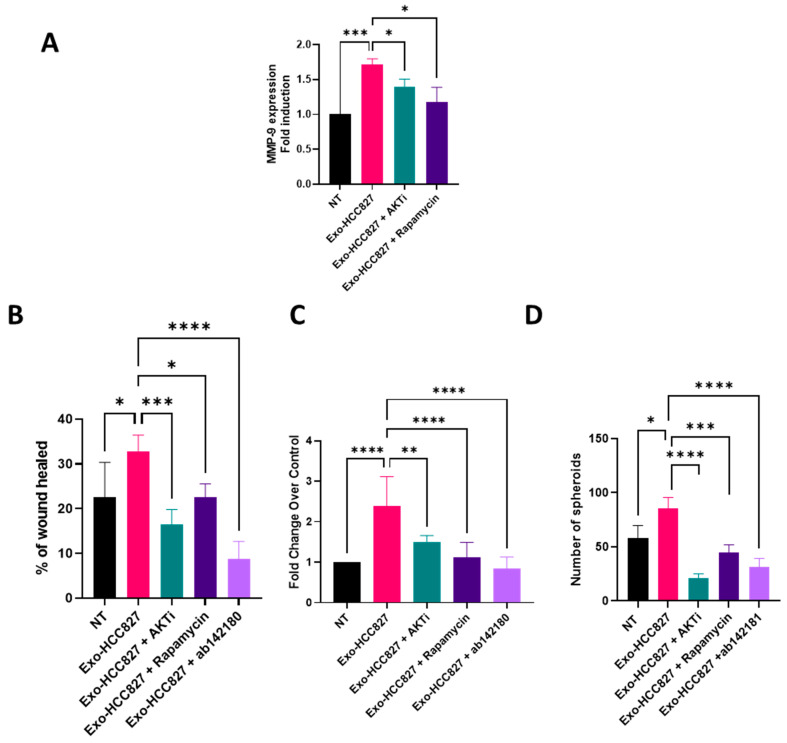
Exosomes from HCC827 cell line induced MMP-9-dependent invasion and migration. (**A**) Quantification analysis of MMP-9 expression obtained by RT-qPCR of protein extracts of A549 cells treated with exosomes isolated from HCC827 supernatant with or without AKTi or rapamycin (mean ± SEM, *n* = 3; statistical analysis was performed using two-way ANOVA). (**B**) Analysis of invasive capacities of A549 cells treated with exosomes isolated from HCC827 supernatant with or without AKTi, rapamycin, or ab142180 by wound healing assay (mean ± SEM, *n* = 3; statistical analysis was performed using two-way ANOVA). (**C**) Analysis of invasive capacities of A549 cells treated with exosomes isolated from HCC827 supernatant with or without AKTi, rapamycin, or ab142180 in a modified Boyden chamber invasion assay (mean ± SEM, *n* = 3; statistical analysis was performed using two-way ANOVA). (**D**) Analysis of 3D outgrowth of A549 cells treated with exosomes isolated from HCC827 supernatant with or without AKTi, rapamycin, or ab142180 by 3D organotypic growth assay (mean ± SEM, *n* = 4; statistical analysis was performed using two-way ANOVA). Significance was set at * *p* < 0.05, ** *p* < 0.01, *** *p* < 0.001 and **** *p* < 0.0001.

**Figure 7 cancers-14-03776-f007:**
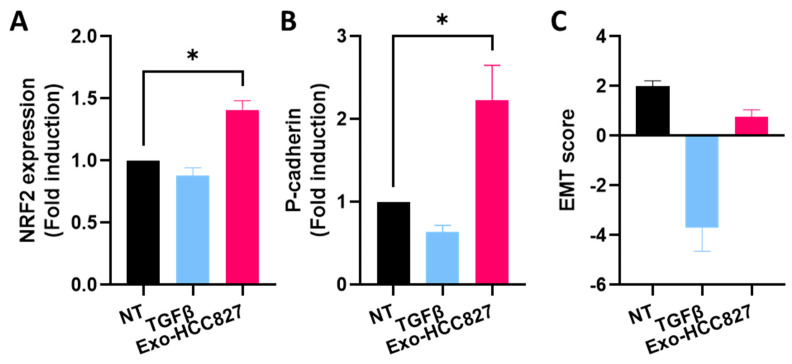
Exosomes from the HCC827 cell line induced hybrid EMT. Quantification analysis of NFR2 (**A**) and P-cadherin (**B**) expression obtained by RT-qPCR of protein extracts of A549 cells treated with TGFβ and exosomes isolated from HCC827 supernatant (mean ± SEM, *n* = 3; statistical analysis was performed using unpaired *t*-test). Significance was set at * *p* < 0.05. (**C**) EMT score calculated by using RT-qPCR levels of two epithelial and three mesenchymal markers. (Chae et al., modified formula).

**Table 1 cancers-14-03776-t001:** Primer sequences.

Gene	Sequence
*Vimentin*	Forward primer: 5′-TGTCCAAATCGATGTGGATGTTTC-3′ Reverse primer: 5′-TTGTACCATTCTTCTGCCTCCTG-3′
*E-cadherin*	Forward primer: 5′-GCTGAGCTGGACAGGGAGGA-3′ Reverse primer: 5′-ATGGGGGCGTTGTCATTCAC-3′
*MMP-9* *(matrix metalloproteinase-9)*	Forward primer: 5′-GGCGCTCATGTACCCTATGT-3′ Reverse primer: 5′-TCAAAGACCGAGTCCAGCTT-3′
*MMP-2* *(matrix metalloproteinase-2)*	Forward primer: 5′-GGCCCTGTCACTCCTGAGAT-3′ Reverse primer: 5′-GGCATCCAGGTTATCGGGGA-3′
*P-cadherin/CDH3*	Forward primer: 5′-AAATGCTCAACCCTGTGTCC-3′ Reverse primer: 5′-ATAGCAACGCAACAGGGAAA-3′
*EPCAM* *(Epithelial cell adhesion molecule)*	Forward primer: 5′-GAAGGCTGAGATAAAGGAGATGGG-3′ Reverse primer:5′-TTAACGATGGAGTCCAA GTTCTGG-3′
*NRF2* *(nuclear factor erythroid 2-related factor 2)*	Forward primer: 5′-CAGCGACGGAAAGAGTATGA-3′ Reverse primer:5′-TGGGCAACCTGGGAGTAG-3′
*GAPDH* *(Glyceraldehyde 3-phosphate dehydrogenase)*	Forward primer: 5′-CCATGTTCGTCATGGGTGTG-3′ Reverse primer: 5′-CAGGGGTGCTAAGCAGTTGG-3′

## Data Availability

The data presented in this study are available in this article (and Appendix A).

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
