# Peer review of "Exosomes from EGFR-Mutated Adenocarcinoma Induce a Hybrid EMT and MMP9-Dependant Tumor Invasion"

_cancers, 2022, doi:10.3390/cancers14153776_

Round 1

Reviewer 1 Report

The manuscript titled “Exosomes from EGFR-mutated adenocarcinoma induce a hybrid EMT and MMP9-dependant tumor invasion” describes the authors want to investigate exosomes derived from EGFR-mutated NSCLC tumor cells promote the progression of cancer via PI3K/AKT/mTOR/MMP-9 signaling. Overall, the manuscript is not well-prepared, especially the figures. Please seriously update the missing information for further consideration. Despite this, the followings are some concerns and comments have been pointed out that the authors may want to consider.

1) Line 80 Materials and Methods section: Please include CAT# and source of reagents in the study to make your study relatively easier reproducible by other researchers.

2) Line 87 Supplementary Table 1: The authors stated “15 NSCLC patients and 5 healthy as control. While in Supplementary Table 1 “Total Clinical stage patients 5+3+5+1=14”, “wild type + EFGR mutation=5+5=10”. Please seriously update the patients’ information.

3) Line 91: Please use subscripted “2” for “CO2”.

4) Line 116: Please include gel concentration.

5) Line 118: Please be consistent with “h” or “hours” throughout the manuscript.

6) Line 120: Please be consistent with or without a space before and after signs throughout the manuscript, for example, “:”.

7) Lines 121-122: “Appropriate secondary antibody” isn’t an accurate academic description.

8) Lines 127-128: Please provide general information on why “A549 cells were treated with exo-HCC827”.

9) Line 130: Please include antibody dilution ratio or concentration.

10) Line 147 immunofluorescence: Please include more details, for example, concentrations and so on.

11) Line 161: Please confirm the images were recorded 6 times per hour for a total of 48 hours. How? Video? There are no videos.

12) Line 179: Please include relatively more details for RT-qPCR.

13) Line 197: I’d suggest the authors use italic p as it refers to a p-value. Check throughout the manuscript.

14) Line 212: Please provide a higher resolution Figure 1.

15) Where are Figure 2 and Figure 3?

16) Figure 4: a) Please provide higher resolution images; b) Please include sample size and statistics information in the figure legend; c) Please define “*” and “**” in the figure legend.

17) Figure 5: Please update the figure legend with more details, A and B.

18) Figure 6: a) Please include sample size and statistics information in the figure legend; b) Please define “*” etc., in the figure legend.

19) Figure 7: a) Please include sample size and statistics information in the figure legend; b) Please define “*” etc., in the figure legend.

20) Are there any western blotting and other results for PI3K/AKT/mTOR/MMP-9 pathway? Please provide.

Author Response

Reviewer #1

R1.1: The manuscript titled “Exosomes from EGFR-mutated adenocarcinoma induce a hybrid EMT and MMP9-dependant tumor invasion” describes the authors want to investigate exosomes derived from EGFR-mutated NSCLC tumor cells promote the progression of cancer via PI3K/AKT/mTOR/MMP-9 signalling. Overall, the manuscript is not well-prepared, especially the figures. Please seriously update the missing information for further consideration. Despite this, the followings are some concerns and comments have been pointed out that the authors may want to consider.

Response: We thank the reviewer for their comments and consideration of this manuscript. We have amended the manuscript accordingly.

R1.1: Line 80 Materials and Methods section: Please include CAT# and source of reagents in the study to make your study relatively easier reproducible by other researchers.

Response: We included catalogue numbers and source for all reagents in this study.

R1.2: Line 87 Supplementary Table 1: The authors stated “15 NSCLC patients and 5 healthy as control. While in Supplementary Table 1 “Total Clinical stage patients 5+3+5+1=14”, “wild type + EFGR mutation=5+5=10”. Please seriously update the patients’ information.

Response: Please see revised Supplementary table 1

R1.3: Line 91: Please use subscripted “2” for “CO2”.

Response: This has been corrected at the specified place. See revised manuscript page 2 line 94.

R1.4: Line 116: Please include gel concentration.

Response: We thank the reviewer for their constructive comments, and we have amended the manuscript accordingly. See revised manuscript page 3 line 119

R1.5: Line 118: Please be consistent with “h” or “hours” throughout the manuscript.

Response: This has been corrected at the specified place and throughout the manuscript.

R1.6: Line 120: Please be consistent with or without a space before and after signs throughout the manuscript, for example, “:”.

Response: We thank the reviewer for their comments, and we have amended the manuscript accordingly.

R1.7: Lines 121-122: “Appropriate secondary antibody” isn’t an accurate academic description.

Response: We thank the reviewer for their constructive comments, and we have amended the manuscript accordingly. See revised manuscript page 3 line 127-128.

R1.8: Lines 127-128: Please provide general information on why “A549 cells were treated with exo-HCC827”.

Response:  A549 cells were  used as they are a well validated cell line for alveolar basal epithelial cells and useful to study epithelial to mesenchymal transition.

R1.9: Line 130: Please include antibody dilution ratio or concentration.

Response: We thank the reviewer for their constructive comments. The company R&D doesn’t provide concentration of their Detection Antibody Cocktails for their kit. We followed manufacturers instruction for dilution.

R1.10: Line 147 immunofluorescence: Please include more details, for example, concentrations and so on.

Response: We thank the reviewer for their constructive comments, and we have amended the immunofluorescence method section in the manuscript accordingly. See revised manuscript page 3 lines 163-165.

R1.11: Line 161: Please confirm the images were recorded 6 times per hour for a total of 48 hours. How? Video? There are no videos.

Response: We thank the reviewer for their comments. A phase contrast image was recorded every 10 min for 48 hours with Nikon Inverted Microscope Incubation Systems Ti-2. This system allows live imaging by keeping the cells in a 5% CO2 humidified environment at 37°C. Please see video in Supplementary video 1.

R1.12: Line 179: Please include relatively more details for RT-qPCR.

Response: We thank the reviewer for their constructive comments, and we have amended the RT-PCR method section in the manuscript accordingly.

R1.13: Line 197: I’d suggest the authors use italic p as it refers to a p-value. Check throughout the manuscript.

Response: This has been corrected at the specified place and throughout the manuscript.

R1.14: Line 212: Please provide a higher resolution Figure 1.

Response: Please see updated Figure 1

R1.15: Where are Figure 2 and Figure 3?

Response: Please see Figure 2 and 3

R1.16: Figure 4: a) Please provide higher resolution images; b) Please include sample size and statistics information in the figure legend; c) Please define “*” and “**” in the figure legend.

Response: We thank the reviewer for their constructive comments, and we have amended the figure accordingly.

R1.17: Figure 5: Please update the figure legend with more details, A and B.

Response: We thank the reviewer for their constructive comments, and we have amended the figure legend accordingly.

R1.18: Figure 6: a) Please include sample size and statistics information in the figure legend; b) Please define “*” etc., in the figure legend.

Response: We thank the reviewer for their constructive comments, and we have amended the figure legend accordingly.

R1.119: Figure 7: a) Please include sample size and statistics information in the figure legend; b) Please define “*” etc., in the figure legend.

Response: We thank the reviewer for their constructive comments, and we have amended the figure legend accordingly.

R1.20: Are there any western blotting and other results for PI3K/AKT/mTOR/MMP-9 pathway? Please provide.

Response: We didn’t perform western blot for PI3K/AKT/mTOR/MMP-9 pathway, the data presented are the results of Phospho-Kinase Array.

Reviewer 2 Report

In this article, Jouida and colleagues, focus their attention on the ability of EGFR mutated- exosomes to promote the epithelial-mesenchymal transition. The overall consideration goes in the direction of a really good paper and I feel that also taking into consideration the below comments the paper could be appreciated by the Cancers readers.

Introduction: A point that could be improved is related to the introductive part of the paper addressing the need for new technologies for the association of a specific marker with an exosome subtype and the exosome subtype to a particular function and/or group of functions s(PMID: 35141731 and others.)
Methods: The methodology section is well described and doesn't need particular revision even if more details in order the exosome purification protocol would be very useful.
Results: The study was well conducted and the results are clearly demonstrated. Figure legends are quite informative and figure resolution looks appropriate. One exception is related to the Panel A Figure 1: please provide a better figure with an improved resolution. The same Panel A Figure 5.

Conclusion and discussion are appropriate.

Good luck.

Author Response

R2.1: In this article, Jouida and colleagues, focus their attention on the ability of EGFR mutated- exosomes to promote the epithelial-mesenchymal transition. The overall consideration goes in the direction of a really good paper and I feel that also taking into consideration the below comments the paper could be appreciated by the Cancers readers.

Response: We thank the reviewer for their consideration, comments and encouragements.

R2.2: Introduction: A point that could be improved is related to the introductive part of the paper addressing the need for new technologies for the association of a specific marker with an exosome subtype and the exosome subtype to a particular function and/or group of functions s(PMID: 35141731 and others.)

Response: We thank the reviewer and have amended the introduction accordingly, please see revised manuscript page 2 line 39-41.

R2.3: Methods: The methodology section is well described and doesn't need particular revision even if more details in order the exosome purification protocol would be very useful.

Response: Further details have been added to the methods section as requested, please see revised manuscript page 2 to 5.

R2.4: Results: The study was well conducted, and the results are clearly demonstrated. Figure legends are quite informative and figure resolution looks appropriate. One exception is related to the Panel A Figure 1: please provide a better figure with an improved resolution. The same Panel A Figure 5. Conclusion and discussion are appropriate. Good luck.

Response: A higher resolution figure 1 has been added and similarly for figure 5.

Reviewer 3 Report

Jouida et al discovered exosomes from serum of patients with EGFR mutated Non-Small Cell Lung Cancer promote the invasion of NSCLC. Overall, the study is well performed, but there are some major points the author would need to address.

Major points:

1. The fonts on all figure labels would need to be enlarged, as the current sizes are too small to be seen clearly.

2. Add one marker (like GAPDH) to show the equivalent loading amounts of protein in fig 1C.

3. Some of the figures are lacking. Where is Figure 3, and why skip Figure 2?

4. Do patients’ EGFR mutation have same mutation as HCC827 (EGFR Del E746-A750 mutation in exon 19)?

5. From Figure 4 showed exosome from HCC827 induce higher level of MMP2 than MMP9. And figure 6 showed exosomes from HCC827 affect the invasion was MMP9 dependent. Need to check MMP2 also.

6. Need to put the discussion of why EGFR mutated of NSCLC can induce the invasion? Though PI3K/AKT signalling?

Author Response

R3: Jouida et al discovered exosomes from serum of patients with EGFR mutated Non-Small Cell Lung Cancer promote the invasion of NSCLC. Overall, the study is well performed, but there are some major points the author would need to address.

Response: We thank the reviewer for their consideration and comments.

R3.1: The fonts on all figure labels would need to be enlarged, as the current sizes are too small to be seen clearly.

Response: Please see updated Figures and Figures legends

R3.2: Add one marker (like GAPDH) to show the equivalent loading amounts of protein in fig 1C.

Response: We performed a western blot to detect the presence or absence of characteristic exosome-associated proteins in order to confirm the presence of exosomes. A loading protein is not necessary. Moreover, the expression level of markers depends on which cell type the exosomes originate from; a protein expressed ubiquitously in exosomes still have not made consensus.

R3.3: Some of the figures are lacking. Where is Figure 3, and why skip Figure 2?

Response: We apologise for this error, please find Figure 2 and 3 in the manuscript.

R3.4: Do patients’ EGFR mutation have same mutation as HCC827 (EGFR Del E746-A750 mutation in exon 19)?

Response: Out of the 5 patients with an EGFR mutation, 4 have an exon 19 deletion and 1 have an exon 21 EGFR L858R mutation.

R3.5: From Figure 4 showed exosome from HCC827 induce higher level of MMP2 than MMP9. And figure 6 showed exosomes from HCC827 affect the invasion was MMP9 dependent. Need to check MMP2 also.

Response: We thank the reviewer for their constructive comments. MMP2 activity and expression is induced by exo-HCC827. Although, its expression doesn’t seem to be PI3K/AKT/mTOR dependant as we observed by RT-PCR (data not shown). Further investigations are needed to determine the implication of MMP-2 in invasion and the signalling cascade involved. Moreover, we focused on MMP-9 as it seems more relevant in adenocarcinoma EGFR mutated than MMP-2; as seen in Figure 3F, there is no significant difference in activity in A549 exposed to exosomes isolated from patient with wild-type adenocarcinoma and with exosomes from patients with EGFR mutation. While MMP-2 would be interesting to investigate, it requires a separate future study.

R3.6: Need to put the discussion of why EGFR mutated of NSCLC can induce the invasion? Though PI3K/AKT signalling?

Response: We thank the reviewer for their constructive comments, we have amended the manuscript accordingly. See page 13 line 402-403.

Round 2

Reviewer 1 Report

Thank you for the revised manuscript. Please consider the following comments and double-check to homogenous the format throughout the manuscript again before publication.  Good luck.

1) Line 260: Please be consistent with or without a space before and after the signs, for example, “±”, “<”, and so on. Check throughout the manuscript.

2) Lines 262-263: Please use italic p and check throughout the manuscript.

3) Line 318-319 Figure 6: The words and figure images are messed-up.

Author Response

Thank you for the revised manuscript. Please consider the following comments and double-check to homogenous the format throughout the manuscript again before publication.  Good luck.

Response: We would like to thank again the reviewer for their comments and consideration of this manuscript. We have amended the manuscript accordingly

1) Line 260: Please be consistent with or without a space before and after the signs, for example, “±”, “<”, and so on. Check throughout the manuscript.

Response: This has been corrected at the specified place and throughout the manuscript.

2) Lines 262-263: Please use italic p and check throughout the manuscript.

Response: This has been corrected at the specified place and throughout the manuscript.

3) Line 318-319 Figure 6: The words and figure images are messed-up.

Response: We thank the reviewer for their comments, and we have corrected the figure and legends accordingly.

Reviewer 3 Report

No extra comments

Author Response

We thank again the reviewer for their comments and consideration of this manuscript.